# UNSUPERVISED SENTENCE EMBEDDING USING DOCUMENT STRUCTURE-BASED CONTEXT

## ABSTRACT

We present a new unsupervised method for learning general-purpose sentence embeddings. Unlike existing methods which rely on local contexts, such as words inside the sentence or immediately neighboring sentences, our method selects, for each target sentence, influential sentences in the entire document based on a document structure. We identify a dependency structure of sentences using metadata or text styles. Furthermore, we propose a novel out-of-vocabulary word handling technique to model many domain-specific terms, which were mostly discarded by existing sentence embedding methods. We validate our model on several tasks showing 30% precision improvement in coreference resolution in a technical domain, and 7.5% accuracy increase in paraphrase detection compared to baselines.

## 1 INTRODUCTION

Distributed representations are ever more leveraged to understand text (Mikolov et al., 2013a;b; Levy & Goldberg, 2014; Pennington et al., 2014). Recently, Kiros et al. (2015) proposed a neural network model, SKIP-THOUGHT, that embeds a sentence without supervision by training the network to predict the next sentence for a given sentence. However, unlike human reading with broader context and structure in mind, the existing approaches focus on a small continuous context of neighboring sentences. These approaches work well on less structured text like movie transcripts, but do not work well on structured documents like encylopedic articles and technical reports.

To better support semantic understanding of such technical documents, we propose a new unsupervised sentence embedding framework to learn general-purpose sentence representations by leveraging long-distance dependencies between sentences in a document. We observe that understanding a sentence often requires understanding of not only the immediate context but more comprehensive context, including the document title, previous paragraphs or even related articles as shown in Figure 1. For instance, all the sentences in the document can be related to the title of the document (1(a)). The first sentence of each item in a list structure can be influenced by the sentence introducing the list (1(b)). Moreover, html documents can contain hyperlinks to provide more information about a certain term (1(c)). With the contexts obtained from document structure, we can connect ransomware with payment (1(a)) and the four hashes with Locky (1(b)).

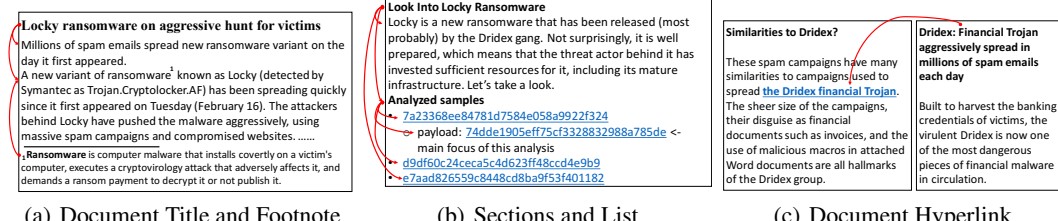

(a) Document Title and Footnote      (b) Sections and List      (c) Document Hyperlink

Figure 1: Examples of long distance dependencies between sentences

Our approach leveraging such structural elements has several advantages. First, it can learn from technical documents containing several subtopics that may cause sudden context changes. Some sentences have dependences to distant ones if a different perspective of the topic is introduced. Using

Table 1: Categorization of sentence embedding methods. * denotes unsupervised methods.

| | | Continuity | |
|---|---|---|---|
| | | Continuous | Discontinuous |
| Range | Intra-sentence | Kusner et al. (2015); Kalchbrenner et al. (2014); Kim (2014); Wieting & Gimpel (2017); Conneau et al. (2017); Palangi et al. (2016); Le & Mikolov (2014)* | Socher et al. (2011); Ma et al. (2015); Tai et al. (2015) |
| | Inter-sentence | Kiros et al. (2015)* | Our work* |

only small neighboring context results in insufficient input to the neural network. Using long distance dependencies, we can provide a broader context. Second, we can consider out-of-vocabulary (OOV) words using their information extracted from the structural context. The vocabulary in a neural network is always limited due to costly training time and memory use. An existing method discarding low frequency words results in losing important keywords in the technical domain.

We validate our model on several NLP tasks using a Wikipedia corpus. When trained with the Wikipedia corpus, our model produces much lower loss than SKIP-THOUGHT in the target sentence prediction task, confirming that training with only local context does not work well for such documents. We also compare the performance of the learned embedding on several NLP tasks including coreference resolution and paraphrase identification. For coreference resolution, our model shows roughly 30% improvement in precision over a state-of-the-art deep learning-based approach on cybersecurity domain, and produces 7.5% increase in accuracy compared with SKIP-THOUGHT for paraphrase identification.

The main contributions of the paper include:

- We propose a general-purpose sentence embedding method which leverages long distance sentence dependencies extracted from the document structure.
- We developed a rule-baed dependency annotator to automatically determine the document structure and extract all governing sentences for each sentence.
- We also present a new OOV handling technique based on the document structure.
- We have applied our methods to several NLP applications using cybersecurity datasets. The experiments show that our model consistently outperform existing methods.

## 2 RELATED WORK

Distributed representation of sentences, which is often called sentence embedding, has gained much attention recently, as word-level representations (Mikolov et al., 2013a;b; Levy & Goldberg, 2014; Pennington et al., 2014) are not sufficient for many sentence-level or document-level tasks, such as machine translation, sentiment analysis and coreference resolution. Recent approaches using neural networks consider some form of dependencies to train the network. Dependencies can be continuous (relating two adjacent words or sentences) or discontinuous (relating two distant words or sentences), and intra-sentence (dependency of words within a sentence) or inter-sentence (dependency between sentences). Many sentence embedding approaches leverage these dependencies of words to combine word embeddings, and can be categorized as shown in 1.

One direct extension of word embedding to sentences is combining words vectors in a continuous context window. Kusner et al. (2015) use a weighted average of the constituent word vectors. Wieting & Gimpel (2017), Conneau et al. (2017), and Palangi et al. (2016) use supervised approaches to train a long short-term memory (LSTM) network that merges word vectors. Kalchbrenner et al. (2014) and Kim (2014) use convolutional neural networks (CNN) over continuous context window to generate sentence representations. Le & Mikolov (2014) include a paragraph vector in the bag of word vectors, and apply a word embedding approaches (Mikolov et al., 2013a;b).

Recently, several researchers have proposed dependency-based embedding methods using a dependency parser to consider discontinuous intra-sentence relationships (Socher et al., 2011; Ma et al., 2015; Tai et al., 2015). Socher et al. (2011) uses recursive neural network to consider discontinuous

dependencies. Ma et al. (2015) proposes a dependency-based convolutional neural network which concatenate a word with its ancestors and siblings based on the dependency tree structure. Tai et al. (2015) proposes tree structured long short-term memory networks. These studies show that dependency-based (discontinuous) networks outperform their sequential (continuous) counterparts.

Unlike these approaches, considering only intra-sentence dependencies, SKIP-THOUGHT (Kiros et al., 2015) joins two recurrent neural networks, encoder and decoder. The encoder combines the words in a sentence into a sentence vector, and the decoder generates the next sentence. Our approach is similar to SKIP-THOUGHT since both approaches are unsupervised and use inter-sentential dependencies. However, SKIP-THOUGHT considers only continuous dependency.

Furthermore, we propose a new method to handle OOV words in sentence embedding based on the position of an OOVword in a sentence and the dependency type of the sentence. To our knowledge, there has been no sentence embedding work incorporating OOV words in formulating the training goal. Most existing systems map all OOV words to a generic *unknown* word token (*i.e.*, $< unk >$). Santos & Zadrozny (2014) and Horn (2017) build an embedding of an OOV word on the fly that can be used as input to our system, but not to set the training goal. Luong et al. (2015) propose a word position-based approach to address the OOV problem for neural machine translation (NMT) systems. Their methods allow a neural machine translation (NMT) system to emit, for each unknown word in the target sentence, the position of the corresponding word in the source sentence. However, their methods are not applicable to sentence embedding, as they rely on an aligned corpus. Also, our approach considers not only word positions but also the dependency types to represent OOV words in a finer-grained OOV level.

## 3 DOCUMENT STRUCTURED-BASED CONTEXT

Previous methods use intra-sentence dependencies such as dependency tree, or immediately neighboring sentences for sentence embedding. However, we identify more semantically related content to a target sentence based on the document structure as shown in Figure 1. In this section, we describe a range of such inter-sentence dependencies that can be utilized for sentence embedding and the techniques to automatically identify them. We use the following notations to describe the extraction of document structure-based context for a given sentence. Suppose we have a document $D = \{S_1, \ldots, S_{|D|}\}$, which is a sequence of sentences. Each sentence $S_i$ is a sequence of words: $s_{i,1}, \ldots, s_{i,|S_i|}$. For each *target sentence* $S_t \in D$, there can be a subset $G \subset D$ that $S_t$ depends on (For simplicity, we use $G$ to denote a $S_t$ specific set). We call such a sentence in $G$ a *governing sentence* of $S_t$, and say $G_i$ governs $S_t$, or $S_t$ depends on $G_i$. Each $G_i$ is associated with $S_t$ through one of the dependency types in $\mathcal{D}$ described below.

### 3.1 TITLES

The title of a document, especially a technical document, contains the gist of the document, and all other sentences support the title in a certain way. For instance, the title of the document can clarify the meaning of a definite noun in the sentence. Section titles play a similar role, but, mostly to the sentences within the section. We detect different levels of titles, starting from the document title to chapter, section and subsection titles. Then, we identify the region in the document which each title governs and incorporate the title in the embedding of all the sentences in the region. To identify titles in a document, we use various information from the metadata and the document content.

**Document Metadata** ($\mathcal{D}_{TM}$)**:** We extract a document title from the <title> tag in a HTML document or from the title field in *Word* or PDF document metadata. Since the document title influences all sentences in a document, we consider a title obtained from $\mathcal{D}_{TM}$ governs every sentence in $D$.

**Heading Tag** ($\mathcal{D}_{THn}$)**:** The heading tags <h1> to <h6> in HTML documents are often used to show document or section titles. We consider all the sentences between a heading tag and the next occurrence of the same level tag are considered under the influence of the title.

**Table Of Contents** ($\mathcal{D}_{TC}$)**:** Many documents contain a table of contents (TOC) providing the overall structure of the document. To detect the titles based on the table of contents, we first recognize a phrase indicating TOC, such as "table of contents", "contents" or "index". Then, we parse the content following the cue phrase and check if it contains a typical TOC pattern such as "Chapter 1 – Introduction" or "Introduction $\cdots\cdots\cdots$ 8". The range of each section can be easily identified

from the TOC. If the document is a HTML file, each line in the TOC tends to have a hyperlink to the corresponding section. For non-HTML documents, we can extract the page number from the TOC (*e.g.*, page 8) and locate the corresponding content if the document includes the page numbers.

**Header and Footer ($\mathcal{D}_{TR}$):** Technical documents often contain the document or section titles in the headers or footers. Thus, if the same text is repeated in the header or in the footer in many pages, we take the text as a title and consider all the sentences appearing in these pages belong to the title.

**Text Styles ($\mathcal{D}_{TS}$):** Titles often have a distinctive text style. They tend to have no period at the end and contain a larger font size, a higher number of *italic* or **bold** text, and a higher ratio of capitalized words compared to non-title sentences. We first build a text style model for sentences appearing in the document body, capturing the three style attributes. If a sentence ends without a period and any dimension of its style model has higher value than that of the text style model, we consider the sentence as a title. Then, we split the document based on the detected titles and treat each slice as a section.

## 3.2 LISTS

Authors often employ a list structure to describe several elements of a subject. These list structures typically state the main concept first, and, then, the supporting points are described in a bulleted, numbered or in-text list as illustrated in Figure 2. In these lists, an item is conceptually more related to the introductory sentence than the other items in the list, but the distance can be long because of other items. Once list items are identified, we consider the sentence appearing prior to the list items as the introductory sentence and assume that it governs all the items in the list.

**Formatted List ($\mathcal{D}_{LF}$):** To extract numbered or bulleted lists, we use the list tags (*e.g.*, <ul>, <ol>, <li>) for HTML documents. For non-HTML documents, we detect a number sequence (*i.e.*, 1, 2, ...) or bullet symbols (*e.g.*, -, ·) repeating in multiple lines.

**In-text List ($\mathcal{D}_{LT}$):** We also identify in-text lists such as "First(ly), …. Second(ly), …. Last(ly), …" by identifying these cue words. We consider the sentence appearing prior to the list items as the introductory sentence and assume that it governs the list items.

---

The categories of the products State Farm offers are as follows:
- We have property and casualty insurance.
- We offer comprehensive types of life and health insurances.
- We have bank products.

---

Figure 2: A sample text with a bulleted list

## 3.3 LINKS

**Hyperlinks ($\mathcal{D}_H$):** Some sentences contain hyperlinks or references to provide additional information or clarify the meaning of the sentence. We can enrich the representation of the sentence using the linked document. In this work, we use the title of the linked document in the embedding of the sentence. Alternatively, we can use the embedding of the linked document.

**Footnotes and In-document Links ($\mathcal{D}_F$):** Footnotes also provide additional information for the target sentence. In an HTML document, such information is usually expressed with in-document hyperlinks, which ends with "#dest". In this case, we identify a sentence marked with "#dest" and add a dependency between the two sentences.

## 3.4 WINDOW-BASED CONTEXT ($\mathcal{D}_{Wn}$):

We also consider the traditional sequential dependency used in previous methods (Kiros et al., 2015; Gan et al., 2017). Given a document $D = \{S_1, \ldots, S_{|D|}\}$, the target sentence $S_t$ is considered to be governed by $n$ sentences prior to ($n < 0$) or following ($n > 0$) $S_t$. In our implementation, we use only one left sentence.

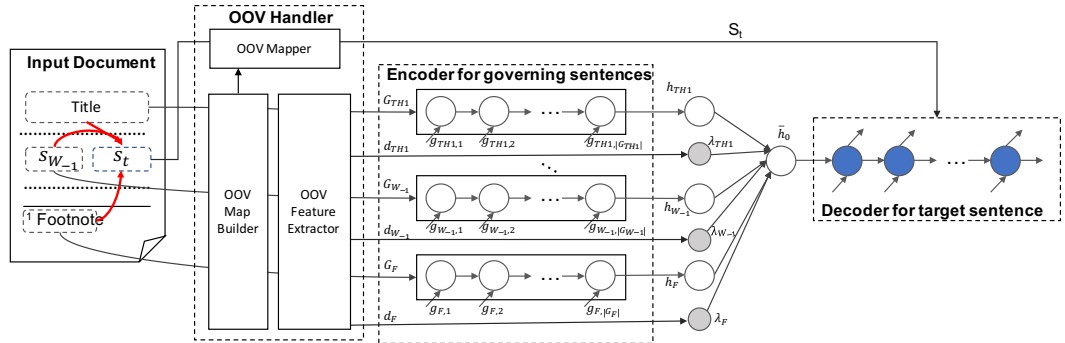

Figure 3: Our model architecture.

## 4 NEURAL NETWORK MODELS

Similarly to SKIP-THOUGHT (Kiros et al., 2015), we train our model to generate a target sentence $S_t$ using a set of governing sentences $G$. However, SKIP-THOUGHT takes into account only the window-based context ($\mathcal{D}_{Wn}$), while our model considers diverse long distance context. Furthermore, we handle out-of-vocabulary (OOV) words based on their occurrences in the context. Our model has several encoders (one encoder for each $G_i \in G$), a decoder and an OOV handler as shown in Figure 3. The input to each cell is a word, represented as a dense vector. In this work, we use the pre-trained vectors from the CBOW model (Mikolov et al., 2013b), and the word vectors can be optionally updated during the training step. Unlike existing sentence embedding methods, which include only a small fraction of words (typically high frequency words) in the vocabulary and map all other words to one OOV word by averaging all word vectors, we introduce a new OOV handler in our model. The OOV handler maps all OOV words appearing in governing sentences to variables and extend the vocabulary with the OOV variables. More details about OOV handler is described in Section 5.

We now formally describe the model given a target sentence $S_t$ and a set $G$ of its governing sentences. We first describe the encoders that digest each $G_i \in G$. Given the $i$-th governing sentence $G_i = (g_{i,1}, \ldots, g_{i,|G_i|})$ let $w(g_{i,t})$ be the word representation (pre-trained or randomly initialized) of word $g_{i,t}$. Then, the following equations define the encoder for $S_i$.

$$h_{i,t} = \text{RC}(w(g_{i,t}), h_{i,t-1}; \theta_E), \quad h_i = h_{i,|G_i|}$$
$$\lambda_i = \sigma(\mathcal{U}d_i + g), \qquad \bar{h}_0 = \sum_i \left\{ \lambda_i(u_{\text{dep}(i)} h_i + a_{\text{dep}(i)}) + (1 - \lambda_i)(W_{\text{dep}(i)} h_i + b) \right\} \quad (1)$$

where RC is a recurrent neural network cell (e.g., LSTM or GRU) that updates the memory $h_{i,t}$; $\theta_E$ is the parameters for the encoder RC; $\lambda_i$ is an OOV weight vector that decides how much we rely on out-of-vocabulary words; $d_i$ denotes the OOV features for $G_i$; $\mathcal{U}$ and $g$ are linear regression parameters; $\sigma(\cdot)$ is the sigmoid function; $u_{\text{dep}}$ and $a_{\text{dep}}$ are dependency-specific weight parameters; $W$ and $b$ are a matrix and a bias for a fully connected layer; and $\bar{h}_0$ is the aggregated information of $G$ and is passed to the decoder for target sentence generation.

Now, we define the decoder as follows:

$$o_t, \bar{h}_t = \text{RC}(o_{t-1}, \bar{h}_{t-1}; \theta_D), \qquad y_t = \text{softmax}(V o_t + c) \quad (2)$$

where RC is a recurrent neural network cell that updates the memory $\bar{h}_t$ and generates the output $o_t$; $\theta_D$ is a set of parameters for the decoder RC; $\text{softmax}(\cdot)$ is the softmax function; and $V o_t + c$ transforms the output into the vocabulary space. That is, $V o_t + c$ generates logits for words in the vocabulary set and is used to predict the words in the target sentence.

To strike a balance between the model accuracy and the training time, we use $K$ randomly chosen governing sentences from $G$ for all target sentence. We use the cross entropy between $y_t$ and $o_t$ as the optimization function and update $\theta_E, W_{\text{dep}(i)}, b, V, c, \theta_D$ and optionally $w(\cdot)$.

---

**Algorithm 1:** Building OOV map

---

**Function** BuildOOVMap $(G, V)$
**Input** : A governing sentence set $G = \{G_1, \ldots, G_{|G|}\}$ and a vocabulary $V_0$
**Output:** OOV Map
**foreach** $G_i \in G$ **do**
$\quad$ OOVWords$_i \leftarrow \{w_j \in G_i | w_j \notin V, j = 1, \ldots, \eta\}$
$\quad$ W2Var$_i \leftarrow \{w_j \rightarrow O_i(j) | w_j \in$ OOVWords$_i, j = 1, \ldots, |$OOVWords$_i|\}$
**return** $\bigcup_i$ W2Var$_i$

---

## 5 OUT-OF-VOCABULARY (OOV) MAPPING

Incorporating all the words from a large text collection in deep learning models is infeasible, since the amounts of memory use and training time will be too costly. Existing sentence embedding techniques reduce the vocabulary size mainly by using only high frequency words and by collapsing all other words to one *unknown* word. The *unknown* word is typically represented by the average vector of all the word vectors in the vocabulary or as a single dimension in a bag-of-word representation. However, this frequency-based filtering can lose many important words including domain-specific words and proper nouns resulting in unsatisfactory results for technical documents.

Specifically, OOV word handling is desired in the following three places: (1) input embeddings to encode the governing sentences ($G$); (2) input embeddings to decode the target sentence ($S_t$); and (3) output logits to compute the loss with respect to $S_t$. In this work, we apply the most commonly used approach, *i.e.*, using the average vector of all the words in the vocabulary to represent all OOV words, to generate the input embeddings of $G$ or $S_t$ for the encoder and the decoder. To handle the OOV words in the output logits, we propose a new method using two vocabulary sets. We first select $N$ most frequent words in the training corpus as an initial vocabulary $V_0$. Note that $N$ (typically, tens of thousands) is much smaller than the vocabulary size in the training corpus (typically, millions or billions). The OOV mapper reduces the OOV words into a smaller vocabulary $V_{OOV}$ of *OOV variables* that can represent certain OOV words given a context (*e.g.*, an OOV variable may indicate the actor in the previous sentence).

We note that only the OOV words appearing in governing sentences influence in model training, and many semantically important words tend to appear in the beginning or at the end of the governing sentences. Thus, we use OOV variables to represent the first and the last $\eta$ OOV words in a governing sentences. Specifically, we denote a $j$-th OOV word in the $i$ dependency governing sentence by an OOV variable $O_i(j) \in V_{OOV}$. This idea of encoding OOV words based on their positions in a sentence is similar to Luong et al. (2015). However, we encode OOV words using the dependency type of the sentence as well as their position in the sentence.

Our OOV handler performs the following steps. First, we build an OOV map to convert OOV words to OOV variables and vice versa. Algorithm 1 summarizes the steps to build a map which converts the first $\eta$ OOV words into OOV variables. To model the last $\eta$ OOV words, we reverse the words in each $G_i$, and index them as $w_{-1}, w_{-2}, \ldots$, then pass them to *BuildOOVMap* to construct $O_i(-1), O_i(-2), \ldots, O_i(-\eta)$.

Note that the mapping between OOV words and OOV variables is many-to-many. For example, suppose "We discuss Keras first" is a target sentence $S_t$, and, "Slim and Keras are two tools you must know" is extracted as the document title by the dependency type $\mathcal{D}_{TS}$, "PLA's weekly review: Slim and Keras are two tools you must know" is extracted as the document title by $\mathcal{D}_{TM}$ for $S_t$, and, words 'Slim', 'Keras' and 'PLA' are OOV words. Then, we map the 'Slim' and 'Keras' from the first title to OOV variable $O_{TS}(1)$ and $O_{TS}(2)$ and 'PLA', 'Slim' and 'Keras' from the second title to $O_{TM}(1)$, $O_{TM}(2)$, and $O_{TM}(3)$ respectively. As a result, 'Keras' in $S_t$ is mapped to $O_{TS}(1)$ and $O_{TM}(3)$.

Once we have the OOV mapping and the augmented vocabulary, we can formulate an optimization goal taking into account the OOV words with a vocabulary with a manageable size. The optimization goal of each RNN cell without OOV words is to predict the next word with one correct answer. In contrast, our model allows multiple correct answers, since an OOV word can be mapped to multiple OOV variables. We use the cross entropy with soft labels as the optimization loss function. The

Table 2: Comparison of our models and SKIP-THOUGHT for target sentence prediction

| Method | All Words | In-vocabulary Words |
|---|---|---|
| OURS | 0.1456 | 0.1394 |
| OURS−DEP | 0.1467 | 0.1415 |
| SKIP-THOUGHT | N/A | 0.1907 |

Table 3: Comparison of paraphrase detection accuracy

| Method | Accuracy |
|---|---|
| OURS | 0.72 |
| SKIP-THOUGHT | 0.67 |

weight of each label is determined by the inverse-square law, *i.e.*, the weight is inversely proportional to the square of the number of words associated with the label. This weighting scheme gives a higher weight to less ambiguous dependency.

One additional component we add related to OOV words is a weight function for the governing sentences based on occurrences of proper nouns ($\lambda_i$ in Equation 1). Instead of equally weighing all governing sentences, we can give a higher weight to sentences with proper nouns, which are more likely to be OOV words. Thus, we introduce a feature vector representing the number of OOV proper nouns in the $i$-th governing sentence ($d_i$ in Equation 1). Currently, the features include # of OOV words whose initials are uppercased, # of OOV words that are uppercased, and # of OOV words with any of the letters are uppercased. Together with the linear regression parameters, $\mathcal{U}$ and $g$, the model learns the weights for different dependency types.

## 6 EXPERIMENTS

In this section, we empirically evaluate our approach on various NLP tasks and compare the results with other existing methods. We trained the proposed model (OURS) and the baseline systems on 807,647 randomly selected documents from the 2009 Wikipedia dump, which is the latest Wikipedia dump in *HTML* format, after removing the discussion and resource (*e.g.*, images) articles among. Since our approach leverages HTML tags to identify document structures, our model use the raw HTML files. For the baseline systems, we provide plain text version of the same articles. All models were train for 300K steps with 64-sized batches and the Adagrad optimizer (Duchi et al., 2011). For the evaluation, we use up-to 8 governing sentences as the context for a target sentence. When a sentence has more than 8 governing sentences, we randomly choose 8 sentences. We set the maximum number of words in a sentence to be 30 and pad each sentence with special start and end of sentence symbols. We set $\eta$ to 4, resulting in $|V_{OOV}| = 80$.

### 6.1 TARGET SENTENCE PREDICTION

Unlike most other approaches, our model and SKIP-THOUGHT (Kiros et al., 2015) can learn application-independent sentence representations without task-specific labels. Both models are trained to predict a target sentence given context. The prediction is a sequence of vectors representing probabilities of words in the target sentence. For a quantitative evaluation between the two models, we compare the prediction losses by using the same loss function, namely cross entropy loss. We randomly chose 640,000 target sentences for evaluation and computed the average loss over the 640K sentences.

We compare SKIP-THOUGHT with two versions of our model. OURS denotes our model using the document structure-based dependencies and the OOV handler. OURS−DEP denotes our model with the OOV handler but using only local context like SKIP-THOUGHT to show the impact of the OOV handler. Table 2 shows the comparison of the three models. The values in the table are the average loss per sentence. We measure the average loss value excluding OOV words for SKIP-THOUGHT, as it cannot handle OOV words. However, for our models, we measure the loss values with and without OOV words. As we can see, both OURS−DEP and OURS significantly outperform SKIP-THOUGHT resulting in 25.8% and 26.9% reduction in the loss values respectively.

## 6.2 PARAPHRASE DETECTION

Further, we compare our model with SKIP-THOUGHT on a paraphrase detection task using the Microsoft Research Paraphrase corpus (Microsoft, 2016). The data consists of 5,801 sentence pairs extracted from news data and their boolean assessments (if the pair of sentences are paraphrases of each other or not), which were determined by three assessors using majority voting. The goal is correctly classifying the boolean assessments and accuracy (# correct pairs / # all pairs) is measured. We used 4,076 pairs for training and 1,725 pairs for testing. Since the data sets contain sentence pairs only and no structural context, we evaluate only the effectiveness of the trained encoder. To compare the qualities of sentence embeddings by the two models, we use the same logistic regression classifier with features based on embedded sentences as in (Kiros et al., 2015). Given a pair of sentences $S_1$ and $S_2$, the features are the two embeddings of $S_1$ and $S_2$, their entry-wise absolute difference, and their entry-wise products. Our model shows a 5% points higher accuracy than SKIP-THOUGHT in paraphrase detection (Table 3), demonstrating the effectiveness of our encoder trained with the structural dependencies. Note that SKIP-THOUGHT trained on the Wikipedia corpus performs worse than a model trained on books or movie scripts due to more sophisticated and less sequential structure in Wikipedia documents.

## 6.3 COREFERENCE RESOLUTION

Traditionally, the coreference resolution problem is considered as a supervised pairwise classification (*i.e.*, mention linking) or clustering problem (coreference cluster identification) relying on an annotated corpus (Haghighi & Klein, 2010; Durrett et al., 2013; Clark & Manning, 2016a;b; Lee et al., 2017). While, recently, there have been an impressive improvement in coreference resolution, existing coreference models are usually trained for general domain entity types (*i.e.*, 'Person', 'Location', 'Organization') and leverage metadata that are not available in technical documents (e.g., *Speaker*). D'Souza & Ng (2015) and Choi et al. (2014) have shown that general domain coreference resolution models do not work well for domain specific entity types.

While our system is not intended to be a coreference resolution tool, the rich sentence embedding can be used for unsupervised coreference resolution allowing it applicable to any domain. Although building a dedicated coreference resolution method to a given domain can produce better results, we claim that our approach can build a good starting set of features without supervision for a new domain. Specifically, we treat the coreference resolution problem as an inference problem given the context. To apply our model, we assume that entity mentions are detected in advance (any mention detection tool can be used), and, for a pronoun or a generic entity reference (e.g., a definite noun phrase), we select a list of candidate referents that conform to the mention types allowed by the pronoun or the definite noun. We apply the mention type-based filtering to reduce the search space, but, a span-based approach as in Lee et al. (2017) can be used as well. Then, we replace the entity reference with each of the candidate referents and compute the loss of the new sentence. Finally, we choose the referent with the lowest loss value as the result, if the ratio of its loss to the original sentence loss value is less than a threshold value $\theta$.

To show the effectiveness of the unsupervised coreference resolution method, we compare our approach with the Stanford Deep Coreference Resolution tool (Clark & Manning, 2016b) using a set of cybersecurity-related documents. The evaluation data consists of 563 coreferences extracted from 38 Wikipedia articles about malware programs which were not included in the training document set. We conducted experiments for several cybersecurity related entity types such as 'Malware' and 'Operating System' in addition to general entity types including 'Person' and 'Organization'. For the evaluation, we set $\theta$ to 0.99 and 1.00.

Table 4 summarizes the results of the two systems. Our model achieves higher precision and recall than DEEPCOREF. Since DEEPCOREF was trained for a general domain, its overall performance on domain specific documents is very low. Figure 4 shows the two systems' performance on different entity types. As we can see, OURS works well for domain specific entities such as 'Malware' and 'Vulnerability', while DEEPCOREF shows higher precision for 'Person' and 'Organization'. The reason OURS performs worse for 'Person' and 'Organization' is because the security documents have only a few mentions about people or organizations, and we did not use carefully crafted features as in DEEPCOREF.

Table 4: Overall performance on coreference resolution

| Method | Precision | Recall |
|---|---|---|
| OURS ($\theta = 0.99$) | 0.47 | 0.12 |
| OURS ($\theta = 1.00$) | 0.35 | 0.17 |
| DEEPCOREF | 0.13 | 0.10 |

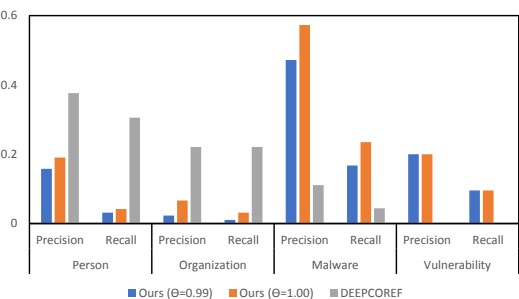

Figure 4: Performance per entity types

### 6.4 DEPENDENCY IMPORTANCE

Note that $u_{dep}$ in Equation 1 denotes the importance level of a dependency $dep$. In Table 5, we show the relative importance of the different dependencies compared to the sequential dependency ($\mathcal{D}_{W,-1}$), which is used in other methods. The values show their relative importance levels when the importance of $\mathcal{D}_{W,-1}$ is set to 1. As we can see, all levels of document and section titles, except the fourth level subsection title, play a much more significant role than the sequential dependency. The reason the title from the metadata, ($\mathcal{D}_{TM}$), does not have a high weight as the title from the heading 1 tag ($\mathcal{D}_{TH1}$) is that the metadata contains extra text, "- Wikipedia", in the title for Wikipedia articles (*e.g.*, "George W. Bush - Wikipedia" instead of "George W. Bush"). Further, hyperlinks ($\mathcal{D}_H$), in-document links ($\mathcal{D}_F$) and formatted lists ($\mathcal{D}_{LF}$) are all shown to have a similar influence as the sequence dependency. The remaining dependencies, $\mathcal{D}_{TC}, \mathcal{D}_{TR}, \mathcal{D}_{TS}$, and $\mathcal{D}_{LT}$ are scarcely found in the Wikipedia corpus, and thus, did not converge or were not updated.

Table 5: The weights of different dependency types. * indicates non converging dependencies.

| $dep$ | $\mathcal{D}_{TM}$ | $\mathcal{D}_{TH1}$ | $\mathcal{D}_{TH2}$ | $\mathcal{D}_{TH3}$ | $\mathcal{D}_{TH4}$ | $\mathcal{D}_{TH5}$ | $\mathcal{D}_{TC}$ |
|---|---|---|---|---|---|---|---|
| $\|u_{dep}/u_{\mathcal{D}_{W,-1}}\|$ | 1.00 | 2.30 | 2.30 | 2.30 | 0.24 | 1.40 | 2.94* |
| $dep$ | $\mathcal{D}_{TR}$ | $\mathcal{D}_{TS}$ | $\mathcal{D}_{LT}$ | $\mathcal{D}_{LF}$ | $\mathcal{D}_H$ | $\mathcal{D}_F$ | |
| $\|u_{dep}/u_{\mathcal{D}_{W,-1}}\|$ | 1.23* | 0.08* | 2.67* | 1.00 | 1.00 | 1.00 | |

## 7 CONCLUSION

In this paper, we presented a novel sentence embedding technique exploiting diverse types of structural contexts and domain-specific OOV words. Our method is unsupervised and application-independent, and it can be applied to various NLP applications. We evaluated the method on several NLP tasks including coreference resolution, paraphrase detection and sentence prediction. The results show that our model consistently outperforms the existing approaches confirming that considering the structural context generates better quality sentence representations.

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
