# OpenReview forum: "UNSUPERVISED SENTENCE EMBEDDING USING DOCUMENT STRUCTURE-BASED CONTEXT"
_ICLR.cc/2018/Conference — Reject_

### Official Review · AnonReviewer3 · 2017-11-26
**Good paper on extracting more signal from document structure**

**Rating:** 7
**Confidence:** 4

**Review:**

This paper presents simple but useful ideas for improving sentence embedding by drawing from more context. The authors build on the skip thought model where a sentence is predicted conditioned on the previous sentence; they posit that one can obtain more information about a sentence from other "governing" sentences in the document such as the title of the document, sentences based on HTML, sentences from table of contents, etc. The way I understand it, previous sentence like in SkipThought provides more local and discourse context for a sentence whereas other governing sentences provide more semantic and global context.

Here are the pros of this paper:
1) Useful contribution in terms of using broader context for embedding a sentence.
2) Novel and simple "trick" for generating OOV words by mapping them to "local" variables and generating those variables.
3) Outperforms SkipThought in evals.

Cons:
1) Coreference eval: No details are provided for how the data was annotated for the coreference task. This is crucial to understanding the reliability of the evaluation as this is a new domain for coreference. Also, the authors should make this dataset available for replicability. Also, why have the authors not used this embedding for eval on standard coreference datasets like OntoNotes. Please clarify.
2) It is not clear to me how the model learns to generate specific OOV variables. Can the authors clarify how does the decoder learns to generate these words.

Clarifications:
1) In section 6.1, what is the performance of skip-thought with the same OOV trick as this paper?
2) What is the exact heuristic in "Text Styles" in section 3.1? Should be stated for replicability.

---

> ### Author Response · Authors · 2018-01-05
> **Thank you for the review.**
>
> - The coreference annotations are done by the two authors of the paper by examining the HTML documents with web browsers. Only nouns/noun phrases and within-document coreferences are considered. We tried to exhaustively annotate the coreferences. We checked all the results of each method to see if a wrong guess is indeed a spurious or we missed it. We are planning to release this evaluation data set publicly.
> - We clarified the procedure to build an OOV mapping in the paper. To explain here a little bit more, OOV variables are defined for each OOV word position for each dependency type. For example, O_TitleMetadata(1) is the first OOV word in the title-metadata governing sentence, which might have a relatively high chance of being the topic word. This OOV variable is later considered mostly in the same way as other words in the decoder. That is, a decoder may output OTitleMetadata(1) and this variable can be mapped back to the OOV word. In our use case, we do not map it back to a word since we focus on embedding not generating a sentence.
> - To evaluate the effect of OOV Handler, we are conducting additional experiments, and we have partial results. If we provide our model only sequential dependencies, while still using OOV Handler, the loss is slightly increased but it is much lower than that of Skip-Thought (reflected in Table 2). The evaluation of OURS - OOV Handler requires re-training of the model due to the change of vocabulary. The training is still ongoing, but the intermediate training loss is much higher than that of OURS at the same number of training iterations (e.g., 24.90 vs 1.58 @ 36160, and 19.77 vs 1.01 @ 61820, each by averaging 832 sentences) and also the reduction rate is lower (21% vs. 36%). We think this shows the importance of OOV handling.
> - Regarding the text style heuristics, we updated the draft to explain it more clearly.

---

### Official Review · AnonReviewer1 · 2017-11-27
**borderline**

**Rating:** 5
**Confidence:** 4

**Review:**

1) This paper proposes a method for learning the sentence representations with sentences dependencies information. It is more like a dependency-based version skip-thought on the sentence level. The idea is interesting to me, but I think this paper still needs some improvements. The introduction and related work part are clear with strong motivations to me. But section 4 and 6 need a lot of details.

2) My comments are as follows:
i) this paper claims that this is a general sentence embedding method, however, from what has been described in section 3, I think this dependency is only defined in HTML format document. What if I only have pure text document without these HTML structure information? So I suggest the authors do not claim that this method is a "general-purpose" sentence embedding model.

ii) The authors do not have any descriptions for Figure 3.  Equation 1 is also very confusing.

iii) The experiments are insufficient in terms of details. How is the loss calculated? How is the detection accuracy calculated?

---

> ### Author Response · Authors · 2018-01-05
> **Thank you for the review.**
>
> Thank you for your thoughtful comments. We tried to address all the concerns you have and revised the paper accordingly.
> - You had a question about generality of the approach. On one hand, to “train” the network, the model requires dependency annotator for each data format (or documents need to be converted to HTML). On the other hand, "embedding a given sentence" does not require dependency or an entire document, as we show in Section 6.2 (paraphrase detection given a pair of sentences without context). To perform a document-level inference such as coreference resolution, we again need formatted documents.
> - We updated the paper to clarify the descriptions of Figure 3 and Equation 1. Also, we discuss more related papers in Section 2, and added explanations about the OOV handler in Section 5.
> - We computed the cross entropy loss for the prediction test. The models predict the next sentence, word by word, and each word is represented as a vector. Each vector is used to compute the cross entropy loss against the corresponding word in the correct next sentence. We clarified the measures, and the experimental set-ups.

---

### Official Review · AnonReviewer2 · 2017-11-28
**A good practical extension of SkipThought, but fairly straightforward**

**Rating:** 5
**Confidence:** 4

**Review:**

This paper extends the idea of forming an unsupervised representation of sentences used in the SkipThought approach by using a broader set of evidence for forming the representation of a sentence. Rather than simply encoding the preceding sentence and then generating the next sentence, the model suggests that a whole bunch of related "sentences" could be encoded, including document title, section title, footnotes, hyperlinked sentences. This is a valid good idea and indeed improves results. The other main new and potentially useful idea is a new idea for handling OOVs in this context where they are represented by positional placeholder variables. This also seems helpful. The paper is able to show markedly better results on paraphrase detection that skipthought and some interesting and perhaps good results on domain-specific coreference resolution.

On the negative side, the model of the paper isn't very excitingly different. It's a fairly straightforward extension of the earlier SkipThought model to a situation where you have multiple generators of related text. There isn't a clear evaluation that shows the utility of the added OOV Handler, since the results with and without that handling aren't comparable. The OOV Handler is also related to positional encoding ideas that have been used in NMT but aren't reference. And the coreference experiment isn't that clearly described nor necessarily that meaningful. Finally, the finding of dependencies between sentences for the multiple generators is done in a rule-based fashion, which is okay and works, but not super neural and exciting.

Other comments:
 - p.3. Another related sentence you could possibly use is first sentence of paragraph related to all other sentences? (Works if people write paragraphs with a "topic sentence" at the beginning.
 - p.5. Notation seemed a bit non-standard. I thought most people use \sigma for a sigmoid (makes sense, right?), whereas you use it for a softmax and use calligraphic S for a sigmoid....
 - p.5. Section 5 suggests the standard way to do OOVs is to average all word vectors. That's one well-know way, but hardly the only way. A trained UNK encoding and use of things like character-level encoders is also quite common.
 - p.6. The basic idea of the OOV encoder seems a good one. In domain specific contexts, you want to be able to refer to and re-use words that appear in related sentences, since they are likely to appear again and you want to be able to generate them. A weakness of this section however is that it makes no reference to related work whatsoever. It seems like there's quite a bit of related work. The idea of using a positional encoding so that you can generate rare words by position has previously been used in NMT, e.g. Luong et al. (Google brain) (ACL 2015). More generally, a now quite common way to handle this problem is to use "pointing" or "copying", which appears in a number of papers. (e.g., Vinyals et al. 2015) and might also have been used here and might be expected to work too.
 - p.7. Why such an old Wikipedia dump? Most people use a more recent one!
 - p.7. The paraphrase results seem good and prove the idea works. It's a shame they don't let you see the usefulness of the OOV model.
 - p.8. For various reasons, the coreference results seem less useful than they could have been, but they do show some value for the technique in the area of domain-specific coreference.

---

> ### Comment · AnonReviewer2 · 2017-12-10
> **More on the coreference experiment**
>
> I was a bit rushed finishing reviews, so here is a longer version of my p.8 coreference experiment thoughts:
>  - While the one cited paper is reasonably representative as a good recent neural coreference paper, it's not the latest, best work, since both Clark & Manning EMNLP 2016 (as opposed to the ACL 2016 paper cited here) and more strongly Kenton Lee et al. (EMNLP 2017) follow and outperform it.
>  - Both of these two papers mainly use learned neural representations and only a handful of handcrafted features (for mention distance, speaker identity, etc.) but argue that they improve performance and hence are still useful. To simply say that you can do it "without handcrafted features" while showing no evidence that the same features would not have improved the performance of your system too isn't clear forward progress. They could run their systems without handcrafted features at the cost of a couple of points in performance too.
>  - You don't say how you "first identify a pronoun or an entity reference". Is this in fact by using a parser and then using handwritten patterns/rules?
>  - You don't say how you find candidate referents that conform to the pronoun type and entity type of the reference. Again, this sounds like handwritten patterns/rules.
>  - Is the dataset you use as a test set available to others, or will it be?
>  - You do show the useful result that a generic newswire supervised learning coref system (Clark and Manning 2016) performs quite poorly on a narrow technical domain (software malware and vulnerabilities), whereas an unsupervised similarity measure can do much better. This is interesting and potentially shows that this work is valuable!
>  - However, looking at the results in Figure 4, your performance on Person and Organization entities is extremely poor, whereas performance of Clark and Manning (2016) is at least far better and moderately good, if not great. So, it doesn't really look like this is a coref system that you can use unsupervised and have it work well.
>  - Overall, your system comes out well ahead because of its far superior performance on Malware and Vulnerabilities. But how much is the failure of Clark and Manning (2016) a failure of coreference, or is it really that their system fails to detect these entities (most malware has weird names). E.g., if your method for mention detection was used followed by Clark and Manning (2016), would Clark and Manning then do much better? It's impossible to tell from the results presented. Perhaps relevant in this context is a recent paper by Berkeley people and colleagues on entity detection on the dark web ( https://evidencebasedsecurity.org/forums/ ). They don't do coreference but their entity detector could have been paired with the Clark and Manning (2016) system. Also, their data is public and may be of interest.
>  - No examples are given, but I suspect that a lot of the cases of malware coreference are just string identity. That is, they keep referring to something as "W97M.Cloud.1" many times. If so, again this is in principle trivial coreference, and the other system is probably mainly failing on mention detection. It would be great if you could give some statistics on what proportion of string identity coreference decisions there are, how many involve pronouns, etc.
>
> So, overall, section 6.3 seems to fall short of being a well-done experiment.

---

> > ### Author Response · Authors · 2017-12-18
> > **Thank you for the comprehensive comments.**
> >
> > We are compiling answers, and conducting additional analyses to support them. We will get back to you soon!

---

> > ### Author Response · Authors · 2018-01-05
> > **Thank you for the review (Part 1)**
> >
> > Thank you very much for the very thoughtful and constructive comments. Here, we tried to clarify unclear parts you pointed out and address all comments. We have updated the paper accordingly.
> >
> > - In particular, there were many questions about the coreference resolution experiments. Regarding the baseline, we found that the specific implementation we compared with is that of Clark & Manning EMNLP 2016 based on https://github.com/clarkkev/deep-coref, and we fixed the citation. We do not claim that our approach can outperform a dedicated coreference resolution method designed for a specific domain with a large training data set. Instead, our focus is more on semantic embedding of sentences which can be used for many NLP tasks,  and we show how our model can be used as a unsupervised coreference resolution tool for technical domains that may not have enough training data. We revised to make this clear in the draft.
> > - You mentioned the coreference candidate generation is unclear. We run a dependency parser and entity recognition pipeline to annotate entities. We also find pronouns or noun phrases requiring coreference resolution using a dictionary. The entity type conformity is done using rules on pronouns (i.e., it, they, … can refer to any type, he/she can only refer to a person) or head noun (e.g., the new ransomware -> Malware class, …). We updated the paper to clarify this.
> > - You had a question about the coreference resolution performance per entity types. The main difference between malware/vulnerability and person/organization cases is the amount of contextual information (or dependency). Since the documents are in the cyber security domain, for people or organizations, their coreference is often syntactically obvious, but there is not much context. Our approach does not consider gender/number agreement and other syntactic features well. Another case we observed is when the entity is not mentioned in governing sentences. We expect to solve this problem by expanding the sequential dependency window or the dependency annotator. Therefore, our method focusing on semantic information does not perform well for entities with little context.
> > - You questioned if a domain-specific mention detection would significantly improve the accuracy of the Stanford Coref system.  While a more rigorous experiment is need to answer the question, we expect that this might not be critical. Poor mention detection would result in poor recall, but not necessarily poor precision. Their system uses a mention detection method  which is described in Raghunathan et al. (2010), and it does find malware names and vulnerability names as noun phrases. Even though,it does not know the entity type, they are considered as candidate antecedents.  But we can still see low precision. Also, according to Clark and Manning (2016 EMNLP and ACL), they train their network using features such as string matching features and distance features as well as word embeddings. They may have different weights in different domains, and even the word embedding may not be available or useful in a domain corpus as discussed in Pilehvar and Collier (2016).

---

> > ### Author Response · Authors · 2018-01-05
> > **Thank you for the review (Part 2)**
> >
> > - Regarding the malware coreference examples, there are many different cases including “the malware”, “the program”, “the worm”, “WannaCry ransomware” (c.f., WannaCry attack), “Wannacry” (different capitalization/spacing), “WannaCrypt” (other nicknames), different malware names by different antivirus companies (e.g., WORM_WCRY.A, Ransom.Wannacry), “it”. In this domain test data, proper noun string identity almost always indicates the same entity. Therefore, we did not include them in our experiments, and we only considered if the strings are different.
> > - Regarding word embedding of OOV words, you mentioned other approaches like character-based encoding. The main reason we did not consider such approaches is that the application area we target is specialty domains like cyber-security, where the semantic meaning of many terms may not come from character level information (e.g., the embedding of WannaCry should be more similar to Locky or other ransomware than “cry” or “wanna”). Character-based encoding is not very helpful while adding the complexity to the model. However, we included Luong et al.'s method and some others in the paper.
> > - We found the paper by Luong et al. (2015) you mentioned is indeed very related to ours. The main difference to Luong et al. (2015) is that they rely only on positions, but in our approach, we have two dimensions to identify a specific unknown word placeholder: position and dependency. We updated the draft to include this paper. “Pointing” (Vinyals et al. 2015) seems to be an interesting approach handling varying vocabulary size. But although their approach can handle varying output vocabulary, encoding, and training the attention mechanism over the whole English vocabulary having millions of words is computationally infeasible (their experiments covers up to 500 symbols in the vocabulary), or they did not consider a hybrid of a fixed vocabulary and varying vocabulary (i.e., OOV) together which is a nontrivial task.

---

> > ### Author Response · Authors · 2018-01-05
> > **Thank you for the review (Part 3)**
> >
> > - You mentioned the paraphrase evaluation does not show the usefulness of the OOV model. That is yes and no. The OOV Handler works in two different situations. The first is when we train the model, where we are given a raw HTML document that we can build dependency-based OOV mapping. The second one is when we apply the model. In a task like coreference resolution where you are given a document, you can leverage and build an OOV mapping, but the paraphrase identification dataset does not provide raw HTML documents that we can annotate the dependencies. In this case, we are applying the model trained with OOV words (thus with more data, especially for a domain corpus), and the encoder understands that we have to carry some information about an OOV word in a given sentence, which gives better sentence embedding. While the effect of this training can be found in the paraphrase detection experiment, we added the coreference resolution experiment to further analyze the use of OOV words.
> > - To help understanding of the components (i.e., OOV Handler and dependencies) of our approach, we are conducting additional experiments and we have partial results. If we provide our model only sequential dependencies, while still using OOV Handler, the loss is slightly increased but it is much lower than that of Skip-Thought (updated in Table 2). The evaluation of OURS - OOV Handler requires re-training of the model due to the change of vocabulary. The training is still ongoing, but the intermediate training loss is much higher than that of OURS at the same number of training iterations (e.g., 24.90 vs 1.58 @ 36160 and 19.77 vs 1.01 @ 61820, each by averaging 832 sentences) and also the reduction rate is lower (21% vs. 36%). We think this shows the importance of OOV handling.
> >
> > - One difficulty of the evaluation was that there is no available task on documents with structures (they are all plain text of sentence sequences, or at most just with a title). Thus, we conducted the coreference resolution task on Wikipedia documents. We are planning to publicly release this evaluation data set.
> >
> > - Other minor comments including the notation of sigmoid and softmax are fixed. Also, there are several reasons we chose an old version of Wikipedia. First, we wanted to have enough new documents to test on. Second, we wanted to build a real-world system that process a raw HTML document. This Wikipedia dump is the last version provided in HTML format. The idea of using the first sentence of a paragraph seems to be valid, and we agree that this can be observed in Wikipedia articles, and will include it in the future experiments.

---

### Decision · Program_Chairs · 2018-01-29
**ICLR 2018 Conference Acceptance Decision**

**Decision:**

Reject

**Comment:**

The paper presents an interesting extension of the SkipThought idea by modeling sentence embeddings using several document-structure related information.  Out of the various kinds of evaluations presented, the coreference results are interesting -- but, they fall short by a bit (as noted by Reviewer 2) because they don't compare with recent work by Kenton Lee et al.  In summary, the idea provides an interesting bit on building sentence embeddings, but the experimental results could have been stronger.